# Sesquiterpene Lactones as Promising Candidates for Cancer Therapy: Focus on Pancreatic Cancer

**DOI:** 10.3390/molecules27113492

**Published:** 2022-05-29

**Authors:** Laura Cecilia Laurella, Nadia Talin Mirakian, Maria Noé Garcia, Daniel Héctor Grasso, Valeria Patricia Sülsen, Daniela Laura Papademetrio

**Affiliations:** 1Instituto de Química y Metabolismo del Fármaco (IQUIMEFA), CONICET-Universidad de Buenos Aires, Junín 956, Piso 2, Buenos Aires CP 1113, Argentina; lclposdoc@gmail.com; 2Facultad de Farmacia y Bioquímica, Universidad de Buenos Aires, Buenos Aires CP 1113, Argentina; nadiatmirakian@gmail.com; 3Cátedra de Inmunología, Departamento de Microbiología, Inmunología, Biotecnología y Genética, Facultad de Farmacia y Bioquímica, Universidad de Buenos Aires, Buenos Aires CP 1113, Argentina; mnoegarcia@gmail.com; 4Instituto de Estudios de la Inmunidad Humoral (IDEHU), CONICET-Universidad de Buenos Aires, Junín 956, Piso 4, Buenos Aires CP 1113, Argentina; dgrasso@ffyb.uba.ar; 5Cátedra de Fisiopatología, Departamento de Ciencias Biológicas, Facultad de Farmacia y Bioquímica, Universidad de Buenos Aires, Buenos Aires CP 1113, Argentina

**Keywords:** pancreatic ductal adenocarcinoma, sesquiterpene lactones, nuclear factor-kB, mitogen-activated protein kinase (MAPK), phosphoinsotide-3 kinase (PI3K)

## Abstract

Pancreatic ductal adenocarcinoma (PDAC) is a highly aggressive disease which confers to patients a poor prognosis at short term. PDAC is the fourth leading cause of death among cancers in the Western world. The rate of new cases of pancreatic cancer (incidence) is 10 per 100,000 but present a 5-year survival of less than 10%, highlighting the poor prognosis of this pathology. Furthermore, 90% of advanced PDAC tumor present KRAS mutations impacting in several oncogenic signaling pathways, many of them associated with cell proliferation and tumor progression. Different combinations of chemotherapeutic agents have been tested over the years without an improvement of significance in its treatment. PDAC remains as one the more challenging biomedical topics thus far. The lack of a proper early diagnosis, the notable mortality statistics and the poor outcome with the available therapies urge the entire scientific community to find novel approaches against PDAC with real improvements in patients’ survival and life quality. Natural compounds have played an important role in the process of discovery and development of new drugs. Among them, terpenoids, such as sesquiterpene lactones, stand out due to their biological activities and pharmacological potential as antitumor agents. In this review, we will describe the sesquiterpene lactones with in vitro and in vivo activity against pancreatic tumor cells. We will also discuss the mechanism of action of the compounds as well as the signaling pathways associated with their activity.

## 1. Introduction

Pancreatic cancer (PC) is one of the most difficult conditions to deal with and a highly important challenge for biomedical research. Nowadays, pancreatic malignancy is a death sentence with poor survival expectancy and without a truly effective treatment option. Moreover, the global incidence of PC will tend to increase over the next 20 years [1], positioning this mortal pathology as a priority in finding novel therapeutic approaches.

Sesquiterpene lactones (STLs) are a group of natural terpenoid compounds that contain fifteen carbon atoms and, in most cases, a cyclic arrangement. They are characteristic of the Asteraceae family, but they can also be found in Apiaceae, Magnoliaceae and Lauraceae, among others. Sesquiterpene lactones are classified into four main groups: germacranolides, eudesmanolides, guaianolides and pseudoguaianolides. The suffix “olide” refers to the presence of a γ-lactone group, which often contains an *exo*-methylene group conjugated to the carbonyl group [2,3].

This type of natural terpenoid has a wide range of biological activities reported with antitumor, anti-inflammatory and antiparasitic activities being the ones that stand out [2]. Several investigations in the last years have focused on the anticancer potential of STLs [3,4]. The in vitro activity of STLs in different tumor cells lines as well as the effect of the compounds on experimental in vivo models has been reported. The anticancer effect of STLs has been associated with the modulation of survival pathways [4,5,6]. Many of the STLs have shown promising activity and selectivity of action. The cytotoxic and antitumor activities and other biological properties of STLs have been attributed to the presence of the α-methylene-γ-lactone, which can react with nucleophiles by a Michael addition. However, other structural features, such as electronic and steric factors, can influence the activity and selectivity of action [4,6,7].

Taking into consideration the incidence of PC and the lack of effective and selective drugs for its treatment and the potential of STLs as anticancer agents, the aim of this review is to provide a revision of the STLs that have demonstrated in vitro or in vivo activity against pancreatic tumor cells. The mode of action of the compounds, as well as the signaling pathways associated to the activity, will also be presented.

## 2. Pancreatic Cancer

The pancreas is a retroperitoneal organ with both endocrine and digestive functions. About 1% of pancreatic tissue corresponds to its endocrine portion represented by the Langerhans’ islets. This endocrine pancreatic function is focused on the glucose metabolism with insulin and glucagon as the main released hormones [8]. The rest of the pancreatic tissue, about 90%, is composed by acini where the pancreas acinar cells synthesize and secrete digestive enzymes that are transported to the duodenum through the interconnected ducts [8]. From these last structures, acini and ducts, the pre-tumoral lesions termed PanINs (pancreas intraepithelial neoplasms) are developed [9]. There are several types of pancreatic cancer, such as neuroendocrine tumors, squamous cell carcinoma and adenosquamous carcinoma, though more than 90% of cases correspond to the pancreatic adenocarcinoma (PDAC) [9]. The initial low-grade PanINs potentially progress with morphological and molecular alterations towards high-grade dysplastic lesions and data support that those tissue lesions can eventually evolve to PDAC [10,11,12]. In contrast to an annual incidence of 13 per 100,000, that is relatively low compared to other malignancies, PDAC possesses a poor 5-year survival of less than 10% which has persisted relatively unchanged over the past 25 years [13,14]. Despite the improvement in the clinical care of PDAC patients, those poor survival rates are explained in part by the aggressiveness of pancreas carcinogenesis and the late diagnosis. Most patients debut with locally advanced or metastatic disease, having a median survival of 6–10 months, and 3–6 months, respectively. Despite 10–15% of patients having potentially resectable tumors, in most cases recurrence of disease follows surgery [15]. It has been revealed that PDAC is preceded by the evolution of precursor lesions, PanINs 1A/B, 2, and 3 [16], and, under particular conditions, acinar ductal metaplasia (ADM) could be critical for the development of PanIN lesions [17]. The evolution from histologically normal ductal epithelium to low-grade PanIN to high-grade PanIN is associated with the accumulation of particular genetic changes, such as K-ras mutations [18]. From a few initial gene mutations pancreatic carcinogenesis progresses with several molecular alterations that include changes in gene expression and the epigenetic profile, gene copy number and chromosomal aberrations [19]. An advanced PDAC tumor possesses genomic instability and plenty of gene alterations. However, there are four of those genes which are constant in all patients: KRAS that is present in more than 90% of tumors and believed to be the initial mutation, followed by TP53, CDKN2A and SMAD4 [13]. Altogether, the genetic profile of PDAC cells with the important influence of tumor microenvironment impacts in several oncogenic signaling pathways, such as RAS/mitogen-activated protein kinase (MAPK) cascade, nuclear factor-kB (NF-κB), phosphoinsotide-3 kinase (PI3K), Jun N-terminal kinase (JNK), transforming growth factor β (TGF-β), WNT-Notch and Hedgehog (HH) among others [20,21].

Less than 20% of PDAC are detected as early and localized tumors, which allow the possibility of a surgical resection. Although resection is one of the best scenarios for a PDAC patient, the 5-year survival rate is still below 31% in those cases [19]. The late diagnosis of PDAC is reflected in the rate of patients presenting metastases mainly in the liver and lungs [13]. Morphologically, the PDAC possesses a high level of heterogeneity with a highly stromal microenvironment that hinders the chemotherapeutics arrival to cancer cells. In addition, the quick environmental adaptability with metabolic reprogramming and metastatic tendency makes the PDAC too tough tumor for the standard therapeutic approaches [19]. Gemcitabine has been the standard treatment for metastatic PDAC for more than 20 years, increasing average survival by only 6.7 months. Combinations have been tested with different compounds, such as 5-fluorouracil (5-FU), folinic acid and nab-paclitaxel, without surprising results. A cocktail of drugs, composed of 5-fluorouracil, leucovorin, oxaliplatin and irinotecan, termed modified FOLFIRINOX and the nab-paclitaxel plus gemcitabine are the most extended chemotherapeutic options, though they have very poor performance with highly toxic effects [22]. FOLFIRINOX provided truly significant improvements in terms of patient survival, increasing to 11.1 months, even in advanced developments, metastatic tumors [23], but with significant adverse effects, much higher than those produced by the gemcitabine which have side effects and lack of efficacy. While the main complications of gemcitabine are high toxicity, damage to healthy tissue and that it has a dose limit to be applied, FOLFIRINOX is associated with an increased risk of infections secondary to neutropenia, reduced cell production of the entire white series, thrombosis and sensory neuropathy [24,25,26]. Therefore, the decision of the therapeutic algorithm to be used lies mainly in the state of health that the patient presents at the time of starting chemotherapy and then, in the evolution of each patient. Although the survival achieved by using FOLFIRINOX is promising, the only effective therapy continues to be surgical resection, but only 9% of patients have locally advanced disease at the time of diagnosis.

Considering all the above mentioned, the PDAC remains one the more challenging biomedical topics thus far. The lack of a proper early diagnosis, the impressive mortality statistics, and the poor outcome with the available therapies, determine that finding novel approaches against PDAC with real improvements in patients’ survival and life quality, is a high priority.

## 3. Activated Survival Pathways in Pancreatic Cancer: KRAS Mutation

Despite the intricacy of PDAC initiation, it has been well established that the KRAS mutation is a key driver of tumor progression. Moreover, several studies report that PDAC is highly dependent on this oncogene for tumor initiation and maintenance [27,28]. The KRAS oncogene encodes for a small GTPase, which acts by exchanging signals from receptors in the plasma membrane to central cellular signaling pathways. The activity of KRAS is regulated by two types of molecules, the GTPase activating proteins (GAPs) and the guanine exchange factors (GEFs), which switch KRAS between inactive (GDP bound) and active (GTP bound) states [29]. In PDAC, the most commonly mutation in KRAS is at the G12 residue. This mutation avoids the interaction of KRAS with GAPs resulting in a KRAS constantly bound to GTP and therefore constitutively activated [30]. This affects the signaling pathways downstream, such as the nuclear transcription factor κB (NF-κB), mitogen-activated protein kinase (MAPK) and phosphoinsotide-3 kinase (PI3K), which regulate several key cellular functions including growth and survival. Unrestrained KRAS signaling results in an increment in the proliferation rate, a decrease in apoptosis trigger and an invasive phenotype [29,30,31] (Figure 1).

### 3.1. NF-κB

GSK3α (glycogen synthase kinase 3 alpha), a key gene in pancreatic carcinoma, is upregulated by KRAS, which leads GSK3α to interact with transforming growth factor-beta activated kinase 1 (TAK1), activating IκB kinase complex (IKK), modulating NF-κB. GSK3β (glycogen synthase kinase 3 beta) also regulates PDAC growth and survival by controlling IKK inhibitors and in consequence, NF-κB signaling [31] (Figure 1, pathway A).

The NF-κB family consists of five proteins: RelA (p65), RelB, c-Rel, p50/p105 and p52/p100. They all share a Rel homology domain (RHD) in their N-terminus, which is essential for dimerization as well as binding to DNA elements. NF-κB activity is regulated by a family of inhibitors called IκB (inhibitor of κB) proteins. IκB proteins hide the nuclear localization signals (NLS) of NF-κB proteins and keep them sequestered in the cytoplasm, in an inactive complex [32]. Two main pathways for NF-κB activation are described: the canonical and non-canonical NF-κB signaling pathways. The canonical NF-κB signaling is induced by pro-inflammatory signals, such as interleukin 1β (IL-1β), interleukin 6 (IL-6), tumor necrosis factor α (TNF-α), pathogen-associated molecular patterns (PAMPs) from microorganisms, agonists for the B- or T-cell antigen receptor (BCR or TCR) and chemicals or radiation [33]. The non-canonical NF-κB signaling pathway is confined to a subset of TNF family members, such as B-cell activating factor (BAFF), CD40L and lymphotoxinβ-(LTβ) [34]. The two pathways of NF-κB activation have several differences, not only the receptors and molecules involved, but also, the triggered response. The signaling pathway induced by the canonical NF-κB pathway initiates the signaling by the connection of IKK with different adaptor molecules. The IKK complex is composed of a regulatory subunit called NEMO and two catalytic ones, known as IKKα and IKKβ. Activated IKK phosphorylates IκB subunits (IκBα, IκBβ and IκBε) inducing the IκB ubiquitination and the consequent proteasomal degradation. The phosphorylation of IκB allows NF-κB release to enter the nucleus and induces the transcription of the different target genes. The responses of the canonical and the non-canonical pathways are different rapidness and transcriptional response. Under physiological conditions, the activation of the canonical NF-κB pathway induces a rapid but transient transcriptional response [33,34]. On the contrary, the activation of the non-canonical NF-κB signaling pathway depends on NIK, which is constantly degraded by an E3 ligase complex, composed by c-IAP1/2 and the adaptor TRAF3/TRAF2, in resting cells. When the receptors BAFF, LTβ and CD40 are activated, the complex TRAF/c-IAP is inactivated and as consequence, NIK is not degraded. In this situation, NIK phosphorylates IKK, which phosphorylates p100/RelB labelling it for proteasomal degradation and the consequent release of p52/RelB, which translocates to the nucleus and induces the transcription of the target genes. Compared to the canonical signaling pathway, non-canonical NF-κB response is delayed, but its transcriptional response is prolonged in time [33,34].

In physiological conditions, NF-κB plays an important role in many cellular activities, such as apoptosis, cell proliferation and survival, and regulates the immune response to inflammation (9–11). However, in PDAC, GSK3α additionally regulates canonical and non-canonical NF-κB signaling and has a key role in inducing drug resistance, meanwhile GSK-3β expression is associated to proliferation and survival [31,35,36,37].

### 3.2. MAPK

One of the results of KRAS mutation is the constant activation of Ras/Raf/MEK/ERK signaling pathways. This dysregulation leads to a disbalance of the regulation of cellular processes, such as growth, proliferation, survival and apoptosis, all features with a major role in oncogenesis [38,39]. The Ras/Raf/MEK/ERK pathway is activated by the interaction of extracellular signals with the cytoplasmic and nuclear effectors, various growth factors, adhesion molecules and differentiation factors mediating the activation of other pathways, such as JAK2/STAT3 and PI3K/Akt [40,41,42] (Figure 1, pathway B).

RAS protein is a small GTPase which is activated after binding with growth factors, cytokines and hormone receptors. RAS behaves as a binary signal switch cycling between ON and OFF states due to binding with guanine nucleotide. In the resting cell, RAS is tightly bound with GDP (guanosine diphosphate), which, after binding of an extracellular stimuli to the appropriate cell membrane receptor, is replaced for GTP (guanosine triphosphate), activating the signaling pathway [43,44]. The RAS protein is responsible not only for post-regulation of Raf, MEK and ERK kinases, but also regulates the activity of JNK, PI3K, Akt, NF-κB and others that further regulate various survival cell functions [43,45,46].

RAS kinase binds with GTP for regulating normal cell growth, but due to the mutations in KRAS, it might result in malignant transformation. RAS is responsible for the activation of RAF which additionally activates mitogen effector kinase and MAPK [47].

MAPK (MEK) can be found in two different forms, MEK1 and MEK2. These kinases when phosphorylated further activate to ERKs proteins (ERK1 and 2). The binding of RAS with RAF leads to the activation of the MAPK–ERK kinases, responsible for cell survival, proliferation, transcription, translation, lipid metabolism and protein acetylation [48].

### 3.3. PI3K

Pancreatic cancer, among others, present alterations in the PI3K pathway, including amplification, mutation, or loss of key regulators [39]. PI3K is a family of lipid enzymes that phosphorylate the 3′-OH group of the inositol ring converting it into phosphatidylinositol (PI) on plasma membrane [49]. This pathway can be activated by signals from both extracellular and intracellular sources, including growth factors and nutrients, leading to downstream signaling implicated in cancer growth, survival and tumor progression [39,50]. Moreover, the PI3K pathway is also essential for many cancer-associated events, including angiogenesis, macrophage transcriptional reprogramming, T cell differentiation and fibroblast-supported chemoresistance [51,52,53,54]. PI3Ks are divided into three classes, based on the primary structure and lipid substrate specificity: class I, II and III (Figure 1, pathway C).

The class I PI3K in the most extensively studied. PI3Ks-Class I are composed of two subunits, one catalytic and the other regulatory. Class I PI3K is subdivided into two subclasses: IA and IB. Class IA PI3Ks consist of a catalytic subunit of 110 KDa that can be α, β or δ, and class IB is composed of the catalytic subunit p110 PI3Kγ [55]. The expression of the different catalytic subunits is not aleatory. p110α and p110β are found ubiquitously expressed, but p110γ and p110δ are mainly expressed in immune cells, including macrophages, neutrophils and lymphocytes [56]. The catalytic subunit of class I PI3Ks dimerizes with a regulatory one, which can be p85α and β or p55γ. Each of these regulatory subunits can bind to phosphotyrosyl residues present on different adaptor proteins or growth factor receptors through their SH2 domain. This binding allows an allosteric change that leads to activation of the complex p85/p110 PI3K [57]. The activation of downstream signaling effectors is mediated by the generation of PIP3 and PI-3,4-P2, as products of p85/p110 activity. Then, PIP3 can be rapidly metabolized, and its actions can be antagonized by lipid phosphatases, such as PTEN (PI3K-lipid phosphatase and tensin homolog deleted on chromosome ten), a key tumor suppressor for the PI3K pathway [58]. The class IB PI3K heterodimer is composed of a catalytic subunit p110γ and a regulatory subunit p101. This complex is exclusively activated by GPCRs due to their interaction with the β/γ subunits of G proteins [59].

Class II PI3Ks are comprised of a monomeric catalytic subunit, PI3KC2α, PI3KC2β and PI3KC2γ [60,61], that preferentially phosphorylates the 3′-OH of phosphatidylinositol or phosphatidylinositol-4-phosphate (PI4P) [62]. There is no regulatory subunit known to this subfamily, but it was described that the class II PI3Ks enzymes can interact with several proteins that can act as adaptors, providing specific functions. PI3KC2α and PI3KC2β include an N-terminal clathrin-binding (CB) site, suggesting a link with clathrin-coated vesicles. The PI3KC2α N-terminal region seems to inhibit kinase activity [63] and has been associated to clathrin-mediated endocytosis [64]. PI3KC2γ protein interactions, on the contrary, have not been studied. Class II PI3Ks have received less research attention to date, thus it is not easy to generalize on their functions. However, it is known that PI3K class II functions are implicated in glucose transport, insulin signaling, channel regulation, cell migration, cortical remodeling, endocytosis and exocytosis [61]. In relation to membrane trafficking roles, PI3KC2α-induced PI-3-P activity appears to lead endosomal trafficking in endocytic recycling [61,65,66], phagosome maturation [67], late steps in exocytosis [61] and autophagy [68,69], whereas PI-3,4-P2 activity has been associated with a central role in clathrin-mediated endocytosis [64].

Class III PI3K is composed by a single catalytic subunit, PIK3C3 or VPS34 (homologue of the yeast vacuolar protein sorting-associated proteins 34) and its regulatory subunit, VPS15. The VPS34/VPS15 complex is a key regulator of the intracellular membrane localization of Vps34 and its activity [70], of endosomal trafficking and of the autophagy mechanism regulated by mTORC [70,71]. The regulation of membrane trafficking by VPS34 has key roles in endosomal protein sorting, endosome–lysosome maturation, autophagosome formation, autophagy flux and cytokinesis [70,72,73]. In this sense, VPS34 can take part in different complexes specifying the synthesis of PI-3-P pools at individual intracellular membrane [70].

Each PI3K class exerts multiple cellular roles as a result of the regulation of distinct phosphoinositide pools. However, as mentioned, the direct roles of the three classes of PI3Ks can be categorized predominantly as acting in cell signaling (class I, II) or membrane trafficking (class II, III). In general, it is important to highlight that there are exceptions to the rule: activation of the class I PI3K suppresses autophagy, via the well-established PI3K-AKT-mTOR (mechanistic target of rapamycin) complex 1 (mTORC1) pathway. In contrast, the class III PI3K catalytic subunit Vps34 forms a protein complex with BECN1 and PIK3R4 and produces PI-3-P, required for the initiation and progression of autophagy mechanism. The class II enzyme seems to be an alternative source of PI-3-P and autophagic initiator [62].

## 4. In Vitro and In Vivo Bioactive Sesquiterpene Lactones with Activity on KRAS Pathways

In the last ten years, the mechanism of action of several sesquiterpene lactones have been associated with the modulation of one of the three pathways upregulated downstream KRAS. In this section we discuss the implication of sesquiterpene lactones in modulation NF-κB, MAPK and PI3k signaling pathways (Table 1).

### 4.1. NF-κB

Constitutive activation of NF-κB signaling has been widely associated with drug resistance and pancreatic tumor progression. Several sesquiterpene lactones have been identified as modulators of NF-κB activity.

The water-soluble analogue of parthenolide (**1**), isolated from *Tanacetum parthenium* (Asteraceae), dimethylaminoparthenolide (DMAPT) (**2**), was extensively studied. The ability of DMAPT to suppress pancreatic cancer was evaluated by Yip-Schneider et al. [74] at doses of 20 and 40 mg/kg per day alone and in combination with celecoxib. DMAPT significantly decreased the size of gross pancreatic cancer at 40 mg/kg. Tumor invasion to adjacent organs and metastasis was not observed in mice that received DMAPT in combination with celecoxib [74].

Gemcitabine is one of the first line chemotherapeutic agents used worldwide for pancreatic cancer treatment [75], although it has some clinical limitations because the high grade of resistance. This scenario is in part explained by the activation of autophagy as a survival pathway. Papademetrio et al. [76] described in 2014 that gemcitabine increased autophagy level in PANC-1 and MIAPaCa-2 cells in vitro and in vivo and proposed that this is one of the reasons by which gemcitabine is unable to induce cell death. Moreover, if autophagy was inhibited, gemcitabine induced apoptosis. Gemcitabine was also associated with promoting NF-κB activity that, as we have exposed, controls the gene expression responsible for tumorigenesis and inflammatory responses, which results in chemotherapeutic resistance. In this sense, Yip-Schneider et al. [77] demonstrated in vivo in transgenic mice, that intervention with DMAPT and sulindac in combination with gemcitabine delayed and prevented progression of premalignant pancreatic lesions in the LSL-KrasG12D; Pdx-1-Cre mouse model of pancreatic cancer. Moreover, these authors associated this effect to the diminishing of NF-κB expression in the mPanINs, in pancreas from the DMAPT/gemcitabine and DMAPT/sulindac/gemcitabine groups [77]. The same year, authors amplified their investigation associating the effects observed by combining DMAPT and gemcitabine. A reduction of tumor size and a decrease in the frequency of liver metastases was described, with the upregulation of NF-κB in immune cells [78].

In 2018, Lamture et al. [79] investigated the effect of the combination of Actinomycin-D (ActD) with DMAPT on the inhibition of PANC-1 cell growth. A drug ratio of DMAPT: ActD (1200:1) was assayed. The researchers proposed that DMAPT inhibits NF-κB pathway and induces depletion of glutathione levels; the last one causing cancer cells to be more susceptible to oxidative stress-induced cell death. Additionally, ActD is a polypeptide antibiotic that binds to DNA and inhibits RNA and protein synthesis by inhibiting RNA polymerase II. Consequently, it was suggested that combining DMAPT and ActD would produce synergistic inhibition of PANC-1 cell growth because DMAPT’s inhibition of NF-κB would enhance induction of apoptosis by ActD, via phosphorylation of c-Jun. The investigation demonstrated that the association of these two drugs induced a higher level of cell death than each drug alone. The studies indicated synergism and moderate synergism at combination concentrations of DMAPT/ActD of 12/0.01 and 18/0.015 μM, respectively [79].

Artemisinin (**3**) and its derivatives have been evaluated against different tumor cell lines. Dihydroartemisinin (DHA) (**4**), a semisynthetic derivative of artemisinin, has been reported to induce apoptosis and suppress the proliferation of pancreatic cells in a concentration dependent manner [80]. DHA downregulated the vascular endothelial growth factor (VEGF) under normoxic conditions, suppressed Bcl-2 and the proliferating cell nuclear antigen (PCNA) and upregulated Bax. Wang et al. [81] reported that this compound also meaningfully decreased the volume of tumors, reduced density of microvessels and downregulated the expression of NF-κB-related proangiogenic gene products [81]. Chen H et al. studied the effect of DHA on pancreatic cancer cells, BxPC3 and AsPC-1, both in vitro and in vivo. They observed that DHA inhibited cell viability, with a downregulation of the expression of proliferating cell nuclear antigen and cyclin D1, and an upregulation of p21. Moreover, apoptosis was detected with a reduction in the Bcl-2/Bax ratio and an increment in the activation of caspase-9, in a dose-dependent manner. In mice bearing BxPC-3 xenograft tumors, administration of DHA prevented tumor growth in a dose-dependent manner, and modulated tumoral gene expression similarly to what they observed in in vitro assays [82,83]. Then, the authors explained that the effects previously observed might be due to inhibition of NF-κB signaling [84].

Wang et al. demonstrated that DHA enhances gemcitabine-induced growth inhibition and apoptosis in two pancreatic cancer cell lines, BxPC-3 and PANC-1. They also investigated the mechanism involved suggesting that the effects are at least partially due to the ability of DHA to deactivate gemcitabine-induced NF-κB activation. DHA highly decreased the expression of its target gene products, such as Bcl-2, Bcl-xL, c-myc and cyclin D1. They also showed, in an in vivo model, that the combination of DHA with gemcitabine induced high levels of apoptosis, as well as decreased Ki-67 index, NF-κB activity and its related gene products, and in consequence, significantly reduced tumor volume [85]. Then, authors deepened their studies showing that the inhibition of NF-κB also occurred in xenograft models [86].

Tumor necrosis factor-related apoptosis-inducing ligand (Apo2L/TRAIL) has been considered as a promising anticancer agent, but in several models, chemoresistance affects its efficacy as a treatment strategy. Kong et al. (2012) studied the efficacy of combining DHA with Apo2L/TRAIL for the treatment of pancreatic cancer. They reported that DHA enhanced the efficacy of Apo2L/TRAIL, observing higher percentages of apoptotic BxPC-3 and PANC-1 cells compared with single-agent treatment in vitro. This effect was accompanied by an increment in the generation of reactive oxygen species and the induction of death receptor 5 (DR5) [87]. Interestingly, DHA not only possess activity against pancreatic tumor cells, but also, favors the expansion of γδ T cells and enhances γδ T cell mediated killing activity against SW1990, BxPC-3 and PANC-1 cancer cells. It is suggested that the effects generated by DHA on γδ T cells were upregulation of intracellular perforin, granzyme B expression and IFN-γ production, revealing the increase in its cytotoxic activity [88].

Jia et al. (2014) investigated the role of autophagy in the apoptosis-induction by DHA. DHA-treated BxPC-3 and PANC-1 cells showed increased levels of LC3-II and upregulated the expression of Beclin-1, characteristic features of autophagy. Moreover, DHA activated the JNK pathway. c-Jun NH₂-terminal kinases (JNKs) are powerfully activated by a stressful cellular environment, such as oxidative stress and chemotherapy [89]. JNK activation was found to depend on ROS resulting from the DHA treatment. Furthermore, it was described that JNK pathway inhibition or Beclin-1 silencing were sufficient to prevent the induction of DHA-induced autophagy. In sum, authors proposed that autophagy can be prompted by DHA treatment through Beclin-1 expression induced by JNK [89].

Monomer and dimer derivatives of DHA, using chalcones as linkers, were synthesized by Gaur et al. (2016). The effect of the compounds was evaluated on tumor cells including MIAPaCa-2 (pancreatic cancer). One of the monomers with a 4,4′-dihydroxychalcone showed the best activity with an IC_50_ value of 22 µM and a selectivity index (SI) of 88 (DHA IC_50_ = 58 µM, SI = 12) [90].

Li et al. investigated the role of RNAs in the response of pancreatic tumor cells to DHA [91]. Through microarray and systematic analysis, they could show that the effects generated by DHA, inhibition of proliferation and angiogenesis and promotion of apoptosis in two different human pancreatic cancer cell lines, were by 5 DHA-regulated microRNAs and 11 of their target mRNAs, involved in these effects via 19 microRNA-mRNA interactions. Authors experimentally verified 4 of these microRNAs, 9 of the mRNAs and 17 of the interactions. Four critical microRNAs (miR-34a-5p, miR-195-5p, miR-30c-5p and miR-130b-3p) were found to regulate the expression of many mRNAs (VEGF, IKKα, Cdk4, Cdk6, MEK1, ERK1, E2F1, E2F3, Rac-1 and CDC42) and their proteins, resulting in being crucial to the anti-pancreatic cancer effects of DHA [91].

The effects of DHA were also investigated in JF-305 cells, a pancreatic cancer cell line of Chinese origin. Li et al. demonstrated that DHA induced inhibition of cell proliferation, increase of ROS, cell cycle arrest in the G2/M phase and apoptosis. Modulation of Bax, Bcl-2, cleaved caspase-3, cleaved caspase-9 and Cyto C levels, downregulation of Bcl-2, upregulation of cleaved caspase-9, Cyto C and Bax, and as consequence, a Bax/Bcl-2 ratio increase, have been described. Therefore, it is proposed that DHA could play their antitumor role by inducing apoptosis of JF-305 cells, and this is possible by the formation and increasing of ROS [92].

Cisplatin (DDP) is a chemotherapy compound used in combination with other drugs or radiotherapy for PDAC therapy. Nevertheless, DDP not only exhibits severe side-effects that can lead to discontinuation of therapy, but tumor cells also acquire drug resistance resulting in serious clinical obstacles. In order to develop a more effective and less toxic therapeutic strategy, Du and collaborators studied the effects of combining DHA with DDP. They found that DHA could strongly improve the cytotoxicity of DDP, reducing its effective concentrations both in vitro and in vivo. Combination of DHA and DDP was found to synergistically inhibit the proliferation, as well as induce DNA damage of PANC1 and SW1990 cells. These processes were substantiated by mitochondrial dysfunction, characterized by altered mitochondrial morphology, diminished respiratory function, reduced ATP production and accumulated mitochondria-derived ROS. Moreover, it was shown that ferroptosis contributed to the cytotoxic effects of DHA and DDP in PDAC cells, with devastating accumulation of free iron and unlimited lipid peroxidation [93].

Artesunate (**5**), another derivative of artemisinin, has exhibited selective activity against PANC-1, BxPC-3 and CFPAC-1 pancreatic cancer cells with IC_50_ values of 26.76, 279.30 and 142.80 μM, respectively. Artesunate-treated cells showed loss of mitochondrial membrane potential and the cell death induction was inhibited in the presence of the reactive oxygen species (ROS) scavenger N-acetylcysteine. Artesunate also generated a tumor regression in an in vivo pancreatic cancer xenografts model. The activity of this compound was comparable with the reference drug gemcitabine [94]. Liu et al. reported the synergistic effects of artesunate, and the diterpene triptolide isolated from *Tripterygium wilfordii* (Celastraceae). They demonstrated that the combination of both compounds could inhibit pancreatic cancer cell growth, and induce apoptosis, accompanied by expression of HSP 20 and HSP 27. Furthermore, the decrease of tumor growth was demonstrated by treatment with the combination of both compounds [95]. In 2019, Wang et al., demonstrated that artesunate induced death in KRAS mutant human pancreatic cancer cells (AsPC-1 and PaTU8988) in a ferroptosis manner, preferable to necrosis or apoptosis. Moreover, this STL also improved the mRNA and protein levels of GRP78 in a concentration-dependent manner in the mentioned cell lines. The knockdown GRP78 increased artesunate-induced ferroptosis both in vitro and in vivo [96].

Li et al. [97] studied the role of NF-κB in the in vitro and in vivo antitumor effects of britanin (**6**) on pancreatic cancer cell lines. Authors demonstrated that the inhibition of cell proliferation and migration and the suppression of tumor progression in a pancreatic cancer xenograft mouse model was via suppression of the NF-κB pathway [97]. Britanin modulated the levels of p50, p65 and P-p65. It was considerably worth noting that the P-p65 protein, which regulates the expression of multiple factors downstream, was importantly decreased in the britanin-treated group. Moreover, the in vivo experiments proved that britanin not only suppressed the tumor progression in a pancreatic cancer xenograft mouse model, but also the compound did not exhibit intolerable toxicity [97].

The potential antitumor effect in pancreatic cancer of deoxyelephantopin (DET) (**7**), a natural sesquiterpene lactone isolated from the Chinese herbal medicine *Elephantopus scaber* (Asteraceae), was evaluated by Ji et al. [98]. The antitumor effects of DET were evaluated alone and in combination with gemcitabine. In vitro experiments revealed that DET suppressed the proliferation, invasion and metastasis of pancreatic cancer cells. This STL induced cell apoptosis via oxidative stress, and improved gemcitabine sensitivity by inhibiting the NF-κB signaling pathway. Furthermore, in vivo experiments displayed that DET not only inhibited pancreatic tumor growth and metastasis but also amplified the antitumor capacity of gemcitabine, which was associated with the downregulation of NF-κB and its downstream gene products [98].

### 4.2. MAPK

Alantolactone (**8**) and isoalantolactone (**9**) are the main bioactive compounds that are present in many medicinal plants, such as *Inula helenium, Inula japonica, Inula racemose* and *Aucklandia lappa* (Asteraceae). These compounds have various pharmacological actions including anti-inflammatory, antimicrobial and anticancer properties, with no significant toxicity. Alantolactone and isoalantolactone has been shown to induce apoptosis by targeting multiple cellular signaling pathways that are frequently deregulated in cancers. Alantolactone has been reported to induce apoptosis on many human cancer cell lines, including PANC-1 cells, among others. Isoalantolactone reduced the mitochondrial potential (ΔΨm) in PANC-1 cells. The treatment with alantolactone and isoalantolactone induced apoptosis by increasing translocation of cytochrome c from mitochondria to cytosol and activation of caspase-3 [99].

Khan et al. described the mechanisms involved in the cytostatic and cytotoxic activity of isoalantolactone [100]. Further mechanistic studies revealed that induction of apoptosis was associated with an increase in the generation of reactive oxygen species, cardiolipin oxidation, reduction of the mitochondrial membrane potential and the release of cytochrome C and cell cycle arrest at S phase. N-acetyl cysteine (NAC), a specific ROS inhibitor, restored cell viability and completely blocked isoalantolactone-mediated apoptosis in PANC-1 cells indicating that ROS are involved in isoalantolactone-mediated apoptosis. Moreover, an increment in the levels of phosphorylated p38 MAPK was observed by treatment with isoalantolactone, which was previously associated with the increment of ROS species generation [101]. Furthermore, Zheng et al. revealed that alantolactone potently inhibited human pancreatic cancer cells and suppressed constitutively activated STAT3 (signal transducer and activator of transcription 3), associated with tumor cell proliferation, survival, tumor invasion and angiogenesis in pancreatic tumors [102,103,104]. On the contrary, alantolactone had little effect on the EGFR (epidermal growth factor receptor) pathway. Additionally, combination of alantolactone and an EGFR inhibitor, erlotinib or afatinib, demonstrated a remarkable synergistic anti-cancer effect against pancreatic cancer cells, further developed as a potential therapy for pancreatic cancer [105].

Yan et al. optimized the extraction method of isoalantolactone from *I. helenium* and obtained a mixture of active ingredients with precise proportions (termed as F35), which were alloalantolactone (**10**), alantolactone (**8**) and isoalantolactone (**9**) at the ratio of 1:5:4 respectively. F35 antitumor activity was compared with isoalantolactone on pancreatic cancer cells. Therefore, F35 exhibited practically the same anti-proliferation activity as isoalantolactone in two cell lines. Equally, F35 and isoalantolactone could induce mitochondrion-related apoptosis at the concentration of 6 µg/mL. Besides, F35 inhibited colony-formation and migration of PANC-1 and SW1990 cells. Lastly, F35 presented comparable anti-proliferation and anti-migration effect as isoalantolactone on two pancreatic cancer cell lines, demonstrating that alantolactone or alloalantolactone might have similar antitumor effect as isoalantolactone [106].

### 4.3. PI3K

Parthenolide (**1**) is an STL isolated from *Tanacetum parthenium* (Asteraceae), commonly named fever-few. The cytotoxic activity of parthenolide and the water-soluble analogue of parthenolide, dimethylaminoparthenolide (DMAPT) (**2**), on different tumor cell lines and their antitumor potential have been reviewed by several research groups. With regards to pancreatic cancer, the effect of parthenolide on three human pancreatic tumor cell lines (BxPC-3, PANC-1 and MIAPaCa-2) was evaluated by Yip-Schneider et al. [107]. Substantial growth inhibition was detected at concentrations between 5 and 10 µM in the three cell lines. It was also demonstrated that this STL increased the amount of the NF-κB inhibitory protein, IκB-A, and decreased NF-κB DNA binding activity. The combination of parthenolide with a nonsteroidal anti-inflammatory drug (sulindac) showed synergistic activity in MIAPaCa-2 and BxPC-3 cells and additive activity in PANC-1 cells. In 2010, Liu et al. demonstrated that parthenolide inhibited the growth and invasion of human pancreatic BxPC-3cells in a dose dependent manner. Parthenolide significantly inhibited the cellular growth and invasion of BxPC-3 cells at 5 and 15 μM. Moreover, induction of apoptosis was demonstrated in parthenolide-treated pancreatic cancer cells, by downregulation of procaspase-3 and Bcl-2 and upregulation of caspase-9 and Bax [108]. Then, in 2017, Liu and collaborators amplified their studies about the mechanisms involved in the antitumor effects they had observed and reported that parthenolide suppressed the growth and induced apoptosis of PANC-1 and BxPC3 cells by modulating cell cycle through the depression of cyclin D1. Moreover, they demonstrated that parthenolide induced autophagy-mediated apoptosis in pancreatic cancer cells with a 50% inhibitory concentration (IC_50_) fluctuating between 7 and 9 μM after 24 h of treatment. Notable autophagy was tempted by parthenolide treatment in pancreatic cancer cells. This STL increased the proportion of autophagic cells and meaningfully increased the expression levels of p62/SQSTM1, Beclin 1 and LC3-II in PANC-1 cells. Additionally, inhibiting autophagy by chloroquine, 3-methyladenine or LC3-II siRNA significantly blocked parthenolide-induced apoptosis, signifying that parthenolide-induced apoptosis is dependent on autophagy [109].

The sesquiterpene lactone 2α-chloro-3β,9β-dihydroxy-1β,10β-epoxy-4α,6αH-guai- 11(13)-en-12,5-olide (**11**) isolated from the 70% ethanolic extract of *Artemisia vulgaris* (Asteraceae) was tested for its preferential cytotoxic activity against the PANC-1 cell line in nutrient-deprived medium (NDM) and standard nutrient-rich medium (Dulbecco’s modified Eagle medium, DMEM). Results displayed moderate preferential cytotoxicity (PC_50_) values of the compound, which represent the concentration at which 50% of cancer cells are killed in NDM, without toxicity in DMEM, suggesting that the PI3K pathway is involved in the response of compound **11** [110].

Moeinifard et al. evaluated the anticancer properties of the sesquiterpene lactone britanin (**6**) isolated from *Inula aucheriana* (Asteraceae)*,* and its possible mechanism of action in human pancreatic cancer cells, focusing on the PI3K pathway. Authors demonstrated that the induction of apoptosis was through decreasing the expression of BCL-2 and increasing the expression of BAX in AsPC-1 and PANC-1 cells. Additionally, they observed that britanin increased reactive oxygen species (ROS) generation in different intracellular sites of pancreatic cancer cells. They revealed that britanin decreased phosphorylated AKT level and induced the nuclear accumulation of FOXO1 as well as upregulation of FOXO-responsive target BIM in both pancreatic cancer cell lines. In conclusion, they showed that britanin was able to induce mitochondrial apoptotic pathway through ROS production and modulation of the AKT-FOXO1 signaling axis in AsPC-1 and PANC-1 human cells [111].

The role of MAPK in the tumor cell responses against isoalantolactone (**9**) has been investigated for several years. Recently, the implication of PI3K pathway on pancreatic adenocarcinoma cells proliferation inhibition and apoptosis induction in vitro and in vivo was demonstrated. According to the investigation, EGF-PI3K-Skp2-Akt signal axis and canonical Wnt pathway are involved in the observed activity [112]. Moreover, He et al. detected that alantolactone caused the accumulation of autophagosomes due to impaired autophagic degradation and substantially inhibited the activity and expression of CTSB/CTSD (cathepsin B/ cathepsin D) proteins when depleted, causing lysosomal dysfunction [103]. Moreover, they observed that alantolactone inhibited the proliferation of pancreatic cancer cells in vitro and in vivo and enhanced the chemosensitivity of pancreatic cancer cells to oxaliplatin. Additionally, a reduction in TFEB (transcription factor EB) levels was a critical event in the apoptosis and cell death caused by alantolactone. Their investigation demonstrated that this STL, which impaired autophagic degradation, was a pharmacological inhibitor of autophagy in pancreatic cancer cells and markedly enhanced the chemosensitivity of pancreatic cancer cells to oxaliplatin [103].

### 4.4. NF-κB, MAPK and PI3K Inhibition

The phytochemical investigation of the aerial parts of *Centaurea deflexa* (Asteraceae), carried out by Chicca et al. [113], led to the identification of four STLs among other compounds. The anti-proliferative activity of two sesquiterpene lactones, aguerin B (**12**) and a newly identified 15-nor-guaianolide (**13**), against human pancreatic and colonic cancer cells was investigated. Of the two compounds, only aguerin B was demonstrated to induce apoptotic cell death, ratifying the role as pro-apoptotic moiety of the α-methylene-γ-lactone ring present in aguerin B but not in the newly identified 15-nor-guaianolide [113]. Moreover, aguerin B showed a better anti-proliferative effect than 5-fluorouracil (5-FU), another drug commonly used as first line in pancreatic anticancer therapy [114]. The IC_50_ values obtained for aguerin B were significantly lower in comparison to those obtained with 5-FU, showing a greater amount (more than twice) of apoptotic cells compared with vehicle-treated cells [113]. Although the mechanism by which aguerin B exerts antitumor effect in pancreatic cancer has not been studied, it has been described that this compound significantly inhibited human breast cancer cell growth in vitro with a persuasive suppression on metastasis via downregulation of Hif-1α, among others [114]. Yokoi et al. (2004) published that at low O_2_ levels, resistance to gemcitabine was exerted mainly via activation of phosphoinositide 3-kinase/protein kinase B (PI3K/Akt) and nuclear-factor-kappa-light-chain enhancer of activated B-cells (NF-κB) signaling pathways and partially through the MAPK(Erk) signaling pathway [115]. Taking in mind this background, aguerin B results in being a great candidate to be profoundly studied in pancreatic cancer, alone or in combination with different effective signaling pathways inhibitors [116].

A summary of the survival pathways and the active sesquiterpene lactones that affect them, are presented in Figure 2.

## 5. Other Bioactive Sesquiterpene Lactones Tested in PDAC Models

Other sesquiterpene lactones have been active in PDAC models (Table 2).

The STL helenalin (**14**) has been proposed to induce cell death and to abolish clonal survival in a vastly apoptosis-resistant Bcl-2 overexpressing Jurkat cell line as well as in two other Bcl-2 overexpressing solid tumor cell lines (mammary MCF-7; pancreatic L3.6pl) [117].

The Hedgehog (Hh) signaling pathway is one of the pathways involved in the proliferation and survival of pancreatic cancer cells. The major effector of pancreatic cancer development, the transcription factor glioma-associated oncogene (Gli), is a key element of the Hh signaling pathway. A recognized therapeutic strategy for pancreatic cancer is inhibiting Gli. In this aspect, Lee et al. examined the regulation of Gli and the expression of its target genes to identify the germacranolide sesquiterpene lactone 2-propenoic acid, 2-methyl-2,3,3a,4,5,8,9,10,11,11a,-decahydro-6,10-bis-(hydroxymethyl)-3-methylene-2-oxocyclodeca (b) furan-4-yl ester (**15**), isolated from *Siegesbeckia glabrescens*, as an inhibitor of the Sonic Hh (Shh) pathway. The study demonstrated that compound 15 inhibited the osteoblast differentiation induced by Shh and the transcriptional activity mediated by Gli homolog 1 (Gli1) in mesenchymal C3H10T1/2 stem cells. Moreover, it has been shown that this compound reduced cancer cell proliferation, associated to suppression of Gli-mediated transcriptional activity in human pancreatic cancer PANC-1 and AsPC-1 cells. In consequence, STL 15 possibly could be a promising candidate for the treatment of Hh/Gli-dependent pancreatic cancer [118].

The cytotoxic activity of tagitinine C (**16**), a germacranolide STL isolated from *Tithonia diversifolia* (Asteraceae)*,* and derivatives synthesized by regioselective Michael addition, was reported by Hang Au et al. The compounds were tested against three human cancer cell lines (breast human cancer cells MCF-7, breast human cancer cells multi-resistant to drugs MCF7-MDR, pancreas cancer cells MiaPaCa-2) and on a normal cell line (HEK-293) as control. Cytotoxic activity of the compounds was tested using the MTS (3-(4,5-dimethylthiazol-2-yl)-5-(3-carboxymethoxyphenyl)-2- (4-sulfophenyl)-2H-tetrazolium) method. Results demonstrated that the most sensitive cell line was the pancreatic one (MiaPaCa-2) as more than 80% of cancerous cells did not survive after 72 h of incubation with the compounds at the concentration tested (1µM). Additionally, the derivatives were more cytotoxic than tagitinine C (22% cell viability for tagitinine C vs. 6.6–21% for the new compounds) [119].

The in vitro cytotoxicity of 8β-angeloyloxy-9α-hydroxy-14-oxo-acanthospermolide (**17**) and five melampolide STLs, uvedalin (**18**), enhydrin (**19**), polymatin B (**20**), sonchifolin (**21**), and fluctuanin (**22**), isolated from the leaves of *Smallanthus sonchifolius* (Asteraceae), commonly known as ‘‘yacon’’, was described by de Ford et al. The in vitro activity was tested on three cancer cell lines—the T-cell acute lymphoblastic leukemia cell line (CCRF-CEM), the doxorubicin resistant T-cell leukemia cell line (CEM-ADR5000) and the pancreatic carcinoma cell line (MIA PaCa-2)—and on peripheral blood mononuclear cells (PBMC) from healthy human subjects. In vitro cytotoxicity was determined by MTT assays. Polymatin B and fluctuanin were the most cytotoxic compounds in all the cell lines tested, whereas uvedalin was the least cytotoxic in CCRF-CEM and MIA-PaCa-2 cells with IC_50_ values of 9.2 and 21.2 µM, respectively. On peripheral blood mononuclear cells (PBMC) the compounds showed IC_50_ values >20 µM. To expand further understanding of how these natural compounds exert their cytotoxic activity, authors also performed mechanistic studies with polymatin B. In this sense, they found that oxidative stress may be involved in the polymatin B mechanism of action. Remarkably, ROS formation largely induced different effects: apoptosis in CCRF-CEM cells, necroptosis in CEM-ADR5000 cells through induction of RIP1K, neither apoptosis nor necroptosis in MIA-PaCa-2 cells. Moreover, cells also died partly by necrosis [120].

From the shoots and roots of *Artemisia indica* (Asteraceae), the STL ludartin (**23**) and other compounds were isolated [121]. Ludartin was assessed against MIAPaCa-2 tumor cell line, among others. This STL showed an IC_50_ value of 31.21 ± 0.5 µM.

Sclareolide (**24**) is an STL isolated from *Salvia sclarea*, *Salvia yosgadensis* and *Cigar tobacco*. Sclareolide was combined with gemcitabine and evaluated against pancreatic cancer cells. It was observed that the association induced cell death through apoptosis, the upregulated expression of hENT1, downregulated expression of RRM1 and inhibition of the EMT through the TWIST1/Slug pathway, which was mediated by NICD/Gli1. The authors proposed the combination as a potential strategy for the treatment of patients with gemcitabine-resistant pancreatic cancer [122].

Zhou et al. reported the isolation of STLs from *Ainsliaea yunnanensis* (Asteraceae). The compounds were tested against different tumor cell lines. The STL gochnatiolide C (**25**) was the only one that was active against PANC-1 cells. This compound inhibited the NLRP3 inflammasome’s activation at the concentration of 10 µM [123].

Parthenin is a pseudoguaianolide STL, that is the major constituent of *Parthenium hysterophorus* (Asteraceae). A parthenin analogue (**26**) was tested against pancreatic adenocarcinoma PANC-1, MIAPaCa-2 and AsPC-1 cells. Compound 26 showed the best activity against PANC-1 cells [124].

Recently, Güçlü et al. recounted that tomentosin (**27**), a natural sesquiterpene lactone found in *Inula* spp., had antitumor properties against pancreatic cancer cell lines. Authors demonstrated that tomentosin inhibited MIAPaCa-2 and PANC-1 cell proliferation with a IC_50_ of 33.93 μM for MIAPaCa-2 cells and 31.11 µM for PANC-1 cells after 48 h. This process was accompanied by reactive oxygen species (ROS) production and mitochondrial membrane potential (MMP) disruption [125].

## 6. Discussion and Conclusions

Sesquiterpene lactones are a group of fifteen-carbon natural terpenoid compounds, that have a variety of reported structures and activities. Most of them have a cyclic arrangement and in the last years they have been pointed out as candidates for the development of new drugs to treat different diseases, including cancer [126]. It has been demonstrated that alkylant properties of sesquiterpene lactones play an important role in the antitumor activity. This type of compound has different targets to promote cell death affecting enzymes and cellular signaling cascades [127]. However, other features, such as the presence double bonds in the structure as well as either epoxide, hydroxyl, keto, ester, or in combination, functional groups, influence their activity.

Sesquiterpene lactones have demonstrated in vitro activity and selectivity against tumor pancreatic cells and have shown activity in in vivo models. Some of them have also demonstrated synergistic activity when combining with antitumor agents and with anti-inflammatory drugs. Chemically, the bioactive compounds belong to the main skeletal types: guaianolides, pseudoguaianolides, eudesmanolides and germacranolides, being the greatest group. However, a high number of the studies have been conducted with artemisinin and its derivatives, which are classified as cadinanolides, and are currently in use as antimalarials. In this sense, the STLs artemisinin and its derivatives, alantolactone, isoalantolactone, alloalantolactone and thapsigargin, which have been included in patents related to pancreatic cancer, deserve to be highlighted [128]. These compounds have shown specific modes of action that could lead to the development of more effective and selective drugs to treat pancreatic cancer. In addition, Moujir et al. [3] emphasized the potential application of dimethylaminoparthenolide, a derivative of parthenolide in combination with gemcitabine or actinomycin-D for the treatment of pancreatic cancer.

As it was described in this review, many STLs with different skeletal types have shown activity either in vitro, in vivo or both, on PDAC. This point suggests that in spite of being the bioactive STLs from different carbocyclic groups, the observed activity could be related to other characteristics, such as molecular geometry, chemical environment and electronic and esteric parameters, among other factors. Additionally, the presence of two or more reactive centers in the structure as well as the orientation of the groups present in the skeleton could affect the activity.

Recent investigations have shown that the antitumor potential of STLs against certain types of leukemia and other tumors could be related to the transcription factor c-Myb and the CCAAT-box/enhancer-binding protein beta (C/EBPβ). The c-myb gene is a proto-oncogene that encodes the transcription factor c-Myb, which plays an important role in the proliferation and differentiation of some tumor cells. The inhibition of c-Myb gene transcription could be an interesting strategy to explore the potential of STLs as anticancer agents. It has also been demonstrated that some STLs, such as helenalin acetate and goyazensolide, inhibit the interaction of C/EBPβ with the co-activator p300, which is needed for the activation of c-Myb transcriptional activity. It has also been described that this transcriptional machinery could be involved with the anti-inflammatory activity of some STLs [127,129]. These recent findings highlight the importance of STLs in the search of new and selective antitumor and anti-inflammatory compounds.

Sesquiterpene lactones have been proposed to affect different survival pathways in PDAC, such as NF-κB, MAPK and PI3K, which impact in cellular functions that determine apoptosis and reduce cellular invasion and metastasis in some cases. Different processes have been associated with apoptotic cell death. Among them, the increment of ROS seems to be a central point. Even more, sesquiterpene lactones can be combined with current chemotherapy, not only improving the cytotoxic effects of each drug alone, but also allowing the use of lower doses, with the obvious consequence of improving the quality of the patient’s life by reducing the side effects. In this sense, several studies have demonstrated the potential of some STLs in combination therapy, and as sensitizing agents to help and enhance the action of drugs in clinical use [3]. Moreover, the obtention of derivatives of STLs could lead to compounds with pharmacological significance since either the pharmacokinetics, stability, potency, selectivity or in combination, could be improved by modification of the natural STL [3,130].

Little is known about the effects of sesquiterpene lactones on tumor cells, specifically on PDAC and the immunomodulation response, despite allergic reactions of STLs due to activation of immune system has been described [131]. It seems to have a stimulating role over cytotoxic immune cells, favoring the elimination of tumor cells. The continued preclinical and clinical studies of these types of compounds could be useful for the development of novel effective and selective therapeutic agents helpful for pancreatic cancer treatment. Beside this, more studies are required to understand the mechanisms involved in the elimination of pancreatic tumor cells, beyond the apoptosis, to deeply investigate the role of other mechanisms of cell death, such as necroptosis and immunogenic cancer cell death, as well as the implication of senescence and autophagy as mechanisms that play a dual role, being tumor suppressors or promoters, depending on the circumstances. These studies will help clarifying the scenario and will contribute towards the development of efficient drugs and combinations therapies.

## Figures and Tables

**Figure 1 molecules-27-03492-f001:**
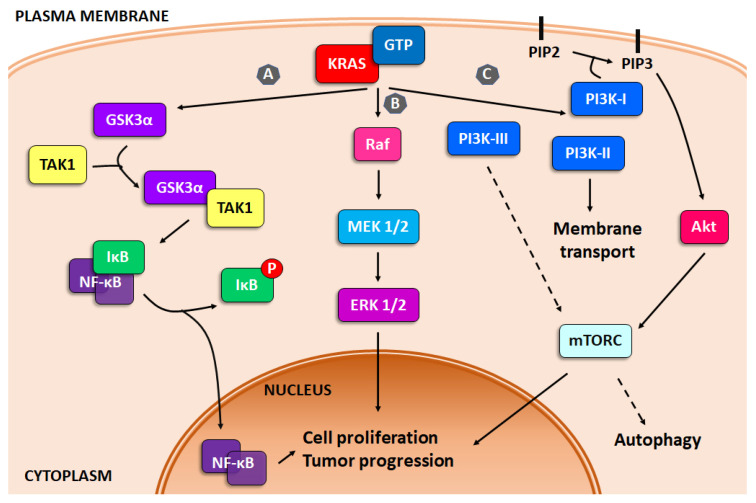
KRAS signaling. Schematic representation of the three main survival pathways upregulated by KRAS mutations. (**A**) NF-κB, (**B**) MAPK and (**C**) PI3K pathways.

**Figure 2 molecules-27-03492-f002:**
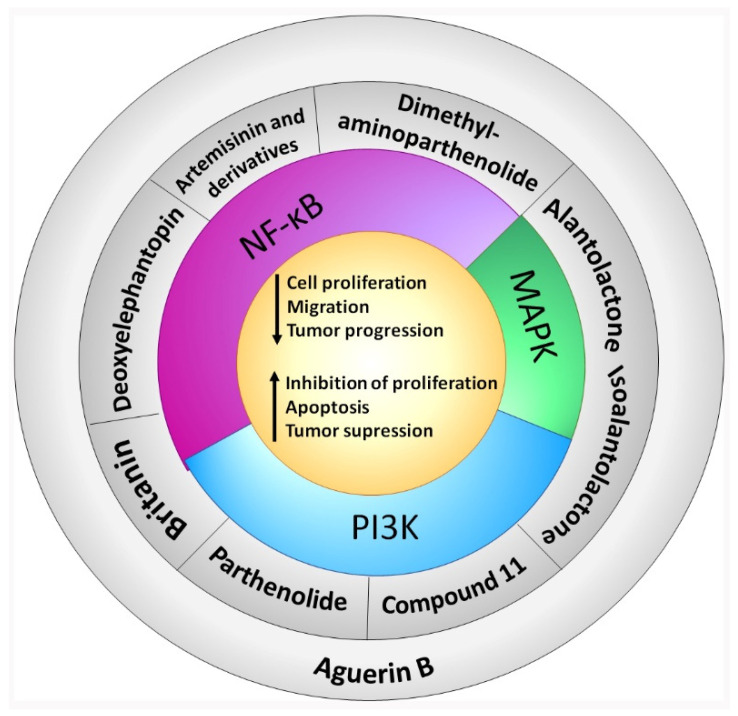
Sesquiterpene lactones and survival pathways. The activity of the main survival pathways upregulated by KRAS mutation are modulated by several sesquiterpene lactones. Deoxyelephantopin, artemisinin and derivatives, and dimethylaminoparthenolide modulate NF-κB. Parthenolide and compound 11 modulate PI3K. Britanin modulates both NF-κB and PI3K. Alantolactone and isoalantolactone modulate MAPK and PI3K. Aguerin B modulates NF-κB, PI3K and MAPK pathways. “Artemisinin and derivatives” refer to artemisinin, dihydroartemisinin and artesunate. “Compound **11**” is “2α-chloro-3β,9β-dihydroxy-1β,10β-epoxy-4α,6αH-guai-11(13)-en-12,5-olide”.

**Table 1 molecules-27-03492-t001:** Bioactive sesquiterpene lactones on KRAS pathways.

Compound Number	Bioactive Compound Name	Chemical Structures
**1**	Parthenolide	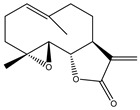
**2**	Dimethylaminoparthenolide (DMAPT)	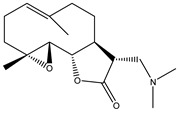
**3**	Artemisinin	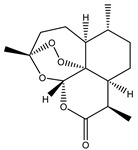
**4**	Dihydroartemisinin (DHA)	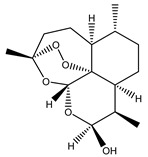
**5**	Artesunate	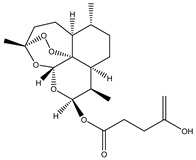
**6**	Britanin	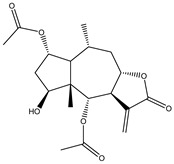
**7**	Deoxyelephantopin (DET)	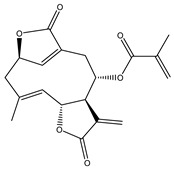
**8**	Alantolactone	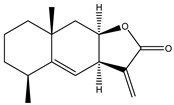
**9**	Isoalantolactone	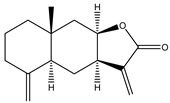
**10**	Alloalantolactone	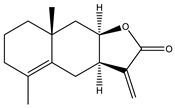
**11**	2α-chloro-3β,9β-dihydroxy-1β,10β-epoxy-4α,6αH-guai-11(13)-en-12,5-olide	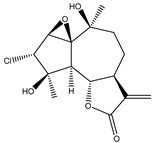
**12**	Aguerin B	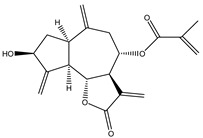
**13**	15-nor-guaianolide	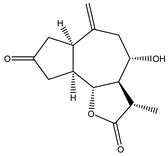

**Table 2 molecules-27-03492-t002:** Bioactive sesquiterpene lactones tested in PDAC models.

Compound Number	Bioactive Compound Name	Chemical Structures
**14**	Helenalin	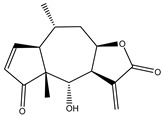
**15**	2-propenoic acid, 2-methyl-2,3,3a,4,5,8,9,10,11,11a,-decahydro-6,10-bis (hydroxymethyl)-3-methylene-2-oxocyclodeca (b) furan-4-yl ester	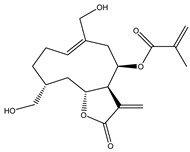
**16**	Tagitinine C	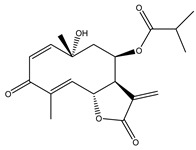
**17**	8β-angeloyloxy-9α-hydroxy-14-oxo-acanthospermolide	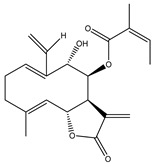
**18**	Uvedalin	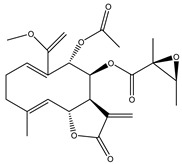
**19**	Enhydrin	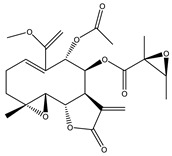
**20**	Polymatin B	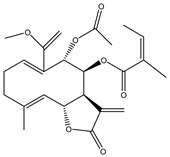
**21**	Sonchifolin	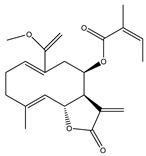
**22**	Fluctuanin	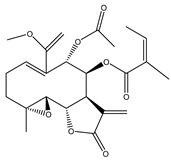
**23**	Ludartin	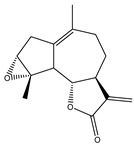
**24**	Sclareolide	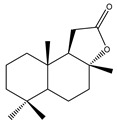
**25**	Gochnatiolide C	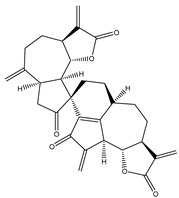
**26**	Parthenin analogue	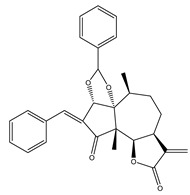
**27**	Tomentosin	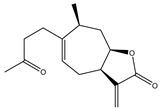

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
