# Peer review of "Sesquiterpene Lactones as Promising Candidates for Cancer Therapy: Focus on Pancreatic Cancer"

_molecules, 2022, doi:10.3390/molecules27113492_

Round 1
Reviewer 1 Report
In the manuscript, the authors reviewed the anticancer effects of sesquiterpene lactones. It also discusses pancreatic ductal adenocarcinoma and its pathogenesis. The work is informative and provides comprehensive insights. However, there are several issues that need to be addressed by the author, summarized below.
Major concern:
1 It’s good to see that the 2D structures of all compounds are clearly represented, but descriptions of the mechanism of action mostly are the conclusions of in vivo or in vitro experiments. The properties of small molecule compounds are mostly determined by their structures, and these properties are the basis of their efficacy. So I suggest adding some discussion about drug efficacy from the perspective of their structures: why could compounds with such skeletons play an anti-PDAC role? And what are the similarities and differences between them?
2 Chemotherapy drugs usually have great side effects, but it is mentioned that the side effects of sesquiterpene lactones are relatively low. It would be better to add some data and discussion to lend color to this view.
3 This manuscript focuses on the anti-PDAC experiments of sesquiterpene lactones, but there is little introduction of existing drugs or therapies for PADC. What are the advantages of sesquiterpene lactones compared with existing drugs?
Minor problems:
1 The structure of the article needs to be optimized. For example, section 2 is slightly repeated in the introduction. Section 4 contains a relatively large amount of content, which alone accounts for half of the manuscript.
Two tables can be added to make the manuscript more readable, especially if several compounds are listed simultaneously.
3 Diagrams describing the signaling pathways mentioned in Section 3 can be added to help readers understand the mechanisms of PDAC.
Reviewer 2 Report
Dear Authors,
This review entitled “Sesquiterpene lactones as promising candidates for cancer therapy: focus on pancreatic cancer” has intensive information in Sesquiterpene lactones and its association with cancer, in particular pancreatic cancer. However, I suggest the following corrections to improve the manuscript:
- Please remove the running title from the head title.
- There are many sentences in this review need to add references. Please review the manuscript and add references for non-cited sentences.
- Please merge the paragraph in line 218 page 5 with the previous paragraph.
- I suggest to make Figure 1 and Figure 3 as a table with writing the name of bioactive compounds of sesquiterpene lactones.
- Please write figure 1 and figure 2 in lines 557 and 558 with a capital letter as Figure 1 and Figure 2.
- I suggest to update the reference list by adding new references published in 2022 if it is possible.
